# Harmony

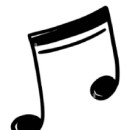

## Platforma e-learningowa do przeprowadzania quizów z harmonii muzycznej z automatycznym sprawdzaniem i generowaniem rozwiązań



**Autorzy**: Przemysław Kojs ⬤ · Mateusz Ejsmont ⬤ · Bartosz Sierzputowski ⬤ · Jakub Wengrzyn ⬤

**Opiekun:** Marek Kopel - K45 Katedra Informatyki Stosowanej

### Streszczenie

Celem projektu jest stworzenie aplikacji wspierającej naukę harmonii muzycznej, która umożliwia użytkownikom tworzenie i rozwiązywanie zadań funkcyjnych w formie quizów.

Aplikacja pozwala na intuicyjne tworzenie quizów z zadaniami funkcyjnymi, które twórcy mogą następnie udostępniać za pomocą linków lub kodów, przypisując je bezpośrednio do pojedynczych użytkowników lub grup, które wcześniej utworzono, a których administratorzy mogą wspomóc twórcę quizu w ocenianiu rozwiązań. Interfejs graficzny do rozwiązywania zadań został zaprojektowany z myślą o intuicyjności i prostocie, umożliwiając wygodne korzystanie ze wszystkich symboli niezbędnych do rozwiązania zadań funkcyjnych. Rozwiązania zadań są automatycznie analizowane przez algorytm oceniający, który sugeruje punktację i identyfikuje błędy w rozwiązaniu, znacznie przyspieszając proces oceniania. System umożliwia uczestnikom także przeglądanie wyników quizów oraz szczegółowych ocen poszczególnych zadań, oferując możliwość udostępnienia im błędów wykrytych przez algorytm oraz możliwość wygenerowania przykładowych poprawnych rozwiązań.

Platforma ta wspiera zarówno uczniów, jak i nauczycieli, usprawniając proces tworzenia i rozwiązywania zadań z harmonii muzycznej oraz automatyzując czynności związane z ocenianiem rozwiązań.

## 1 OPIS PROJEKTU

Nasz projekt koncentruje się na zadaniach funkcyjnych z harmonii. Kazimierz Sikorski, jeden z twórców opracowań związanych z tym tematem, o nauce harmonii wypowiedział się w następujący sposób:

> "Ze wszystkich działów teorii muzyki, nauka harmonii jest niewątpliwie jednym z najważniejszych i najobszerniejszych działów, dotyczy bowiem najbardziej istotnego i **skomplikowanego** elementu muzyki artystycznej, tj. tego, który przejawia się we współbrzmieniach dźwięków i w następstwach tych współbrzmień po sobie." [9]

Celowo uwydatnione zostało słowo "skomplikowany" gdyż jest to element, który mamy na celu wyeliminować.

Tworzenie, rozwiązywanie i sprawdzanie zadań w tradycyjnej formie papierowej niesie za sobą wiele trudności i problemów takich jak:

· Popełnienie błędu podczas rozwiązywania zadania przez ucznia wiąże się z koniecznością przekreślenia i nanoszenia poprawek, co skutkuje nieczytelnością zarówno podczas dalszego rozwiązywania, jak i podczas sprawdzania przez nauczyciela.

· W związku z mnogością reguł i zasad występujących w tej dziedzinie, sprawdzanie kilkunastu rozwiązań przez pedagoga wymaga bardzo dużej uwagi oraz nakładu czasowego, często liczonego w tygodniach.

· Podczas nauki, uczniowie napotykają trudności w samodzielnym sprawdzeniu poprawności swoich rozwiązań, ze względu na małą dostępność zrozumiałych materiałów dydaktycznych, trudność w interpretacji zasad oraz mnogość możliwych właściwych rozwiązań.

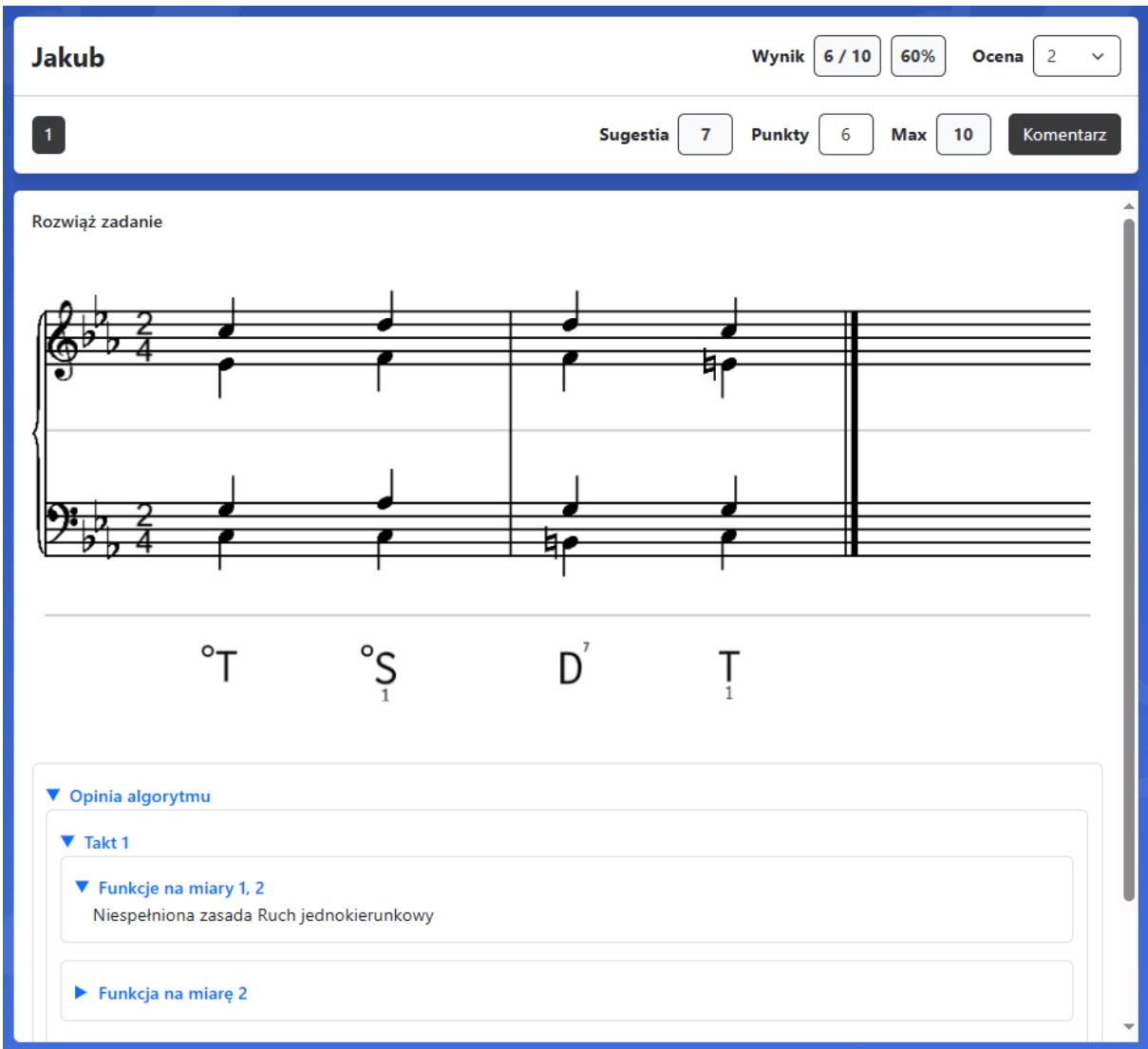

Rysunek 1: Fragment ekranu oceniania zadania, przedstawiający rozwiązanie ucznia (nuty na pięcioliniach), treść zadania (litery pod pięcioliniami) oraz sugerowaną punktację wraz z opisem błędów (pole "sugestia" oraz "Opinia algorytmu" pod rozwiązaniem)

Dzięki cyfryzacji procesu tworzenia i rozwiązywania zadań na intuicyjnych interfejsach graficznych, pozbywamy się problemu ręcznego rysowania i nanoszenia poprawek na papierze. Sprawia to, że są one bardziej czytelne i szybsze do wypełnienia. Wprowadzamy również algorytm sprawdzający rozwiązania, który nieporównywalnie wspomaga i przyspiesza proces oceniania. Na rysunku 1 przedstawiony został ekran oceniania rozwiązania, który obrazuje wspomniane funkcjonalności.

Ponadto umożliwiamy uczniom samodzielne tworzenie zadań, które następnie mogą rozwiązać, a algorytm sprawdzi ich poprawność bez konieczności konsultacji z nauczycielem. Dodatkowo udostępniamy możliwość generowania przykładowych, zawsze poprawnych rozwiązań zadania, co pozwala na porównanie swoich rozwiązań z poprawnymi. Dzięki temu - wraz z algorytmem oceniającym - uczeń w znacznie bardziej przystępny sposób może samodzielnie rozwijać się w tej skomplikowanej dziedzinie.

## 1.1  Prace związane z tematem

### 1.1.1  Analiza istniejących rozwiązań

- **Musition** [1] - aplikacja webowa stworzona przez amerykańską firmę RisingSoftware, która oferuje modułowe nauczanie ogólnych zagadnień muzycznych, nie tylko harmonii. Nauczyciele mogą tworzyć kartkówki przypisane do poszczególnych uczniów. Platforma oparta jest o amerykański system muzyczny, różny od europejskiego.

  Nasz system jest przede wszystkim dostosowany do tutejszego systemu harmonii. Ponadto kładziemy nacisk na możliwość samodzielnej nauki, gdzie uczeń nie jest uzależniony od zadań przy-

pisanych przez nauczyciela, lecz sam może tworzyć i rozwiązywać zadania. Dodatkowo nasza aplikacja oferuje automatyczne sprawdzanie i generowanie rozwiązań zadań.

- **Hooktheory** [2] - aplikacja webowa, która koncentruje się głównie na tworzeniu muzyki i nauce teorii muzycznej. Oferuje interaktywne lekcje harmonii w formie uzupełniania luk w zapisie muzycznym.

  Nasza aplikacja jest znacznie bardziej nastawiona na naukę, w porównaniu do Hooktheory, które duży nacisk kładzie na tworzenie muzyki. Dodatkowo, forma rozwiązywania zadań poprzez wypełnianie luk w quasi-studyjnym edytorze MIDI jest gorzej przystosowana do rozwiązywania zadań z harmonii, co w naszej aplikacji odbywa się z użyciem edytora nut.

### 1.1.2 Technologie

Projekt postanowiliśmy wykonać przy użyciu sprawdzonych technologii firmy Microsoft, które zapewniają wysoką jakość i wydajność. Przede wszystkim, korzystamy z ASP.NET Core [3] wraz z Razor Pages. Zdecydowaliśmy się na to rozwiązanie gdyż jest nam ono dobrze znane, świetnie sprawdza się w budowaniu złożonych aplikacji oraz oferuje wiele bibliotek, takich jak m.in. Entity Framework [5] - do mapowania obiektowo relacyjnego i pracy z bazą danych czy Identity Framework [6] - do autoryzacji i uwierzytelniania. Dane przechowujemy w bazie danych MS SQL Server [7]. Docelowo aplikacja będzie udostępniona na platformie chmurowej Azure [4].

Istotnym aspektem była również decyzja w jaki sposób i z użyciem jakich technologii zostanie zaimplementowany graficzny edytor pięciolinii. Tutaj wybór padł na bibliotekę p5.js [8], która jest bardzo dobrze udokumentowana i w prosty sposób pozwala manipulować elementem HTML `<canvas>`, przy pomocy JavaScript.

### 1.1.3 Ograniczenia projektowe

W związku z ograniczeniem czasowym na realizację projektu, musieliśmy ograniczyć algorytm oceniający rozwiązania do mniej zaawansowanych zadań, ze względu na złożoność i liczbę obowiązujących zasad w dziedzinie problemowej. Ponadto skupiamy się jedynie na zadaniach funkcyjnych z harmonii oraz korzystamy z typowo europejskiego podejścia do zasad.

### 1.1.4 Problemy

W trakcie realizacji projektu zmierzyliśmy się z kilkoma problemami:

- Wydajność edytora graficznego - dla długich i złożonych zadań, edytor graficzny zaczynał mieć problemy wydajnościowe. Po analizie ustaliliśmy, że jest to wina bardzo powolnego wyświetlania się tekstu, obecnego w edytorze. Okazało się, że jest to znany problem tej biblioteki, na który istnieje proste rozwiązanie.

- Publikacja aplikacji na platformie Azure - w końcowych fazach implementacji projektu, gdy nadszedł czas na publikację aplikacji w jej docelowym środowisku produkcyjnym, napotkaliśmy sporo trudności aby ją pomyślnie uruchomić. Wynikało to z wielu elementów konfiguracyjnych, które należało dostosować, aby ostatecznie się to udało.

## 1.2 Wyniki

### 1.2.1 Zaimplementowane funkcjonalności

Udało nam się zaimplementować wszystkie zaplanowane funkcjonalności, które zostały przez nas skrupulatnie przetestowane. Poniżej zostały szczegółowo opisane wszystkie funkcjonalności, których obecność pozwala na wdrożenie projektu w środowisku produkcyjnym:

1. **Uwierzytelnianie użytkowników**

   - Możliwość rejestracji za pomocą e-maila i hasła lub konta Google.
   - Wysyłanie wiadomości z linkiem weryfikacyjnym do potwierdzenia adresu e-mail.
   - Funkcjonalność resetowania hasła poprzez e-mail.

2. **Tworzenie i edycja quizu**

   - Możliwość tworzenia quizów zawierających jedno lub więcej zadań funkcyjnych.
   - Określanie maksymalnej liczby punktów i polecenia do każdego zadania.

- Przechowywanie quizów jako szkiców przed udostępnieniem.
- Edytowanie quizów i ich zadań przed udostępnieniem.
- Możliwość parametrycznego generowania przykładowych zadań.

3. **Tworzenie grup użytkowników**

- Łączenie użytkowników w grupy, które następnie można przypisywać do wypełnienia quizu.
- Dodawanie administratorów grup, którzy mogą zarządzać członkami grupy i oceniać quizy grupy.
- Możliwość zaakceptowania lub odrzucania zaproszenia do grupy.

4. **Udostępnianie quizu**

- Określenie daty otwarcia i zamknięcia quizu, pomiędzy którymi użytkownicy mogą rozwiązywać quiz.
- Udostępnianie quizu użytkownikom za pomocą kodu dostępu, linku lub bezpośrednie przypisanie użytkowników na podstawie e-maila.
- Możliwość przypisania całych grup użytkowników do quizu.
- Nadanie administratorom przypisanych grup uprawnień do oceniania rozwiązań quizów.

5. **Rozwiązywanie quizu**

- Dostęp do przypisanych quizów i ich rozwiązywanie w określonym czasie.
- Możliwość edycji odpowiedzi przed zakończeniem quizu i jego oceną.
- Intuicyjny interfejs graficzny z pełnym zestawem symboli potrzebnych do rozwiązania zadań.

6. **Ocenianie quizu**

- Ocenianie rozwiązań uczestników quizu.
- Sugestie punktowe i wykryte błędy od algorytmu oceniającego.
- Możliwość nadpisania sugestii algorytmu własną oceną punktową i dodania komentarza do poszczególnych zadań.
- Wgląd w sumę punktów za zadania z quizu i możliwość oceny całego quizu.
- Możliwość udostępnienia uczestnikom wykrytych błędów przez algorytm.

7. **Przeglądanie ocen z quizu**

- Możliwość przeglądania ocen rozwiązanych quizów.
- Wgląd w szczegółowe wyniki zadań wraz z komentarzem i ogólną ocenę quizu.
- Możliwość wygenerowania przykładowego poprawnego rozwiązania zadania dla porównania.

### 1.2.2 Osiągnięte cele biznesowe

Dzięki wyżej wspomnianym zaimplementowanym funkcjonalnościom, projekt osiągnął założone cele biznesowe, które wspierają zarówno uczniów w procesie nauki harmonii jak i nauczycieli w procesie tworzenia, udostępniania i oceniania zadań. Poniżej przedstawiono szczegółowy opis osiągniętych celów:

1. **Usprawnienie procesu rozwiązywania zadań:** Intuicyjny edytor nut do rozwiązywania zadań pozwala uczniom na zwinne wypełnianie rozwiązań co pozwala skupić się na poprawności rozwiązania a nie samych symboli muzycznych. Dodatkowo dzięki możliwości nanoszenia poprawek, pozbywamy się nieczytelności z rozwiązań, co wpływa zarówno na pracę ucznia podczas rozwiązywania jak i na pracę nauczyciela podczas oceniania.

2. **Ułatwienie samodzielnej nauki harmonii:** Dzięki umożliwieni uczniom samodzielnego tworzenia i rozwiązywania zadań, mają oni również pełny dostęp do algorytmu oceniającego jak i generowanego rozwiązania. Pozwala im to na ewaluację swoich rozwiązań, wgląd w popełnione błędy oraz uzyskanie wskazówek jak poprawić swoje rozwiązanie. Znacząco ułatwia to samodzielną naukę i zwiększa zaangażowanie uczniów.

3. **Automatyzacja procesu oceniania:** Zaimplementowany przez nas algorytm oceniający, oferujący automatyczną punktację i opis błędów popełnionych w rozwiązaniach zadań, znacząco zmniejsza obciążenie czasowe nauczycieli, pozwalając skupić im się w większym stopniu na dydaktyce przedmiotu a nie ocenianiu zadań. Dodatkowo zapewnia on spójność i obiektywność ocen wszystkich uczniów, nie odbierając jednak możliwości samodzielnej oceny zadań przez nauczyciela według własnych kryteriów.

4. **Usprawnienie współpracy między uczniami a nauczycielami:** Dzięki cyfryzacji procesu tworzenia quizów z harmonii i centralizacji tego w jednym systemie, nauczyciele w prosty i szybki sposób mogą tworzyć i udostępniać zadania nie tylko pojedynczym uczniom ale całym ich grupom. Uczniowie z kolei, w jednym miejscu, mają dostęp do wszystkich przypisanych im zadań oraz wglądu w wystawione oceny wraz z komentarzami i opisami.

### 1.2.3 Wydajność algorytmu oceniającego i generującego rozwiązania

W przeprowadzanych przez nas testach na rzeczywistych zadaniach funkcyjnych o różnych długościach i złożonościach, udało nam się osiągnąć bardzo dobre wyniki pod względem czasu oceniania zadań jak i ich generowaniu przez algorytm. W przypadku oceniania, algorytm zwraca rezultaty już po około 50 ms, niezależnie od długości zadania. Jeśli chodzi o generowanie poprawnych rozwiązań, trwa to poniżej 25 ms, również niezależnie od długości zadania. Takie wyniki są bardzo zadowalające, znacznie przekraczające nasze oczekiwania na ten aspekt systemu. Dzięki takim rezultatom, użytkownik końcowy będzie otrzymywał wsparcie algorytmu, praktycznie bez żadnego widocznego opóźnienia czasowego.

### 1.2.4 Potencjalne zastosowanie projektu

Projekt w obecnej formie może przede wszystkim zostać zastosowany jako platforma do przeprowadzania kartkówek, sprawdzianów czy egzaminów w zakresie zadań funkcyjnych w szkołach lub akademiach muzycznych, a także wszystkich innych placówkach uczących tego działu harmonii. Dzięki możliwości tworzenia grup, przykładowo, nauczyciel mógłby stworzyć grupę zawierającą uczniów jednej klasy, następnie utworzyć dla nich quiz wraz z zadaniami do rozwiązania. Uczniowie mają wówczas określony czas na rozwiązanie quizu, a następnie nauczyciel może z pomocą algorytmu oceniającego sprawnie ocenić rozwiązania uczniów i zwrócić im wyniki. System może również być narzędziem sugerowanym przez nauczycieli jako sposób na samodzielne ćwiczenie zagadnień związanych z tym obszarem muzyki.

Dodatkowo system może być używany przez indywidualnych użytkowników do samodzielnego ćwiczenia zadań z funkcyjnych z harmonii, bez konieczności udziału placówki edukacyjnej w tym procesie, jak to ma miejsce na niektórych platformach.

## 2  WNIOSKI

Projekt został ukończony z pełnią zaplanowanych funkcjonalności, realizując wszystkie zaplanowane cele. Nie tylko wnosi ogromne usprawnienia w kwestii przeprowadzania i oceniania kartkówek czy sprawdzianów przez nauczycieli placówek muzycznych, ale również w znaczny sposób ułatwia i uatrakcyjnia samodzielną naukę osób chętnych rozwijać się w dziedzinie harmonii.

### 2.1  Najważniejszy sukces

Najważniejszym sukcesem projektu było udostępnienie wersji testowej systemu uczniom i nauczycielom Państwowej Szkoły Muzycznej Podwale oraz studentom i wykładowcom Akademii Muzycznej im Karola Lipińskiego we Wrocławiu. W testach wzięło udział kilkanaście osób, otrzymane od nich opinie w kwestii zarówno intuicyjności jak i funkcjonalności systemu były bardzo pozytywne, co potwierdza użyteczność naszego rozwiązania. Dostaliśmy również wskazówki dotyczące potencjalnych miejsc wymagających poprawy, które szczegółowo przeanalizowaliśmy i w większości zaimplementowaliśmy. Ze względu na ograniczenia czasowe, niektórych elementów nie udało nam się poprawić w związku z tym część z nich została przedstawione poniżej. Formularz, który wykorzystywaliśmy do gromadzenia opinii można znaleźć pod adresem.

### 2.2  Kierunki rozwoju

Projekt posiada potencjał do dalszego rozwoju w wielu kierunkach, które mogą uczynić go bardziej wszechstronnym i użytecznym:

· **Zaawansowany algorytm oceniający:** Rozbudowa algorytmu oceniającego o bardziej zaawansowane reguły harmonii, umożliwiając sprawdzanie bardziej skomplikowanych zadań. Można również rozważyć wprowadzenie algorytmów opartych na sztucznej inteligencji, które dostosowywałyby ocenianie do specyficznych wymagań nauczyciela lub analizowałyby różnorodność poprawnych odpowiedzi.

· **Dostosowanie interfejsu pod urządzenia mobilne:** W obecnej wersji aplikacja nie jest w pełni przystosowana pod urządzenia mobilne. Ekrany, na których występuje graficzny interfejs pięciolinii, ze względu na skomplikowany proces dostosowania pod urządzenia mobilne są na nich obecnie mało czytelne. W kolejnych wersjach aplikacji z pewnością jest to element do poprawy. Umożliwienie korzystania z aplikacji na telefonach z pewnością przyczyniłoby się do przyciągnięcia większej liczby użytkowników oraz pozwoliłoby na korzystanie z systemu nie tylko sprzed ekranu laptopa lub komputera. Jest to również element najczęściej występujący w opiniach testerów, w związku z tym jego implementacja w przyszłości byłaby priorytetowa.

· **Rozszerzenie na inne rodzaje zadań muzycznych:** Możliwość dodawania pozostałych typów zadań z zakresu harmonii muzycznej, takich jak analizowanie akordów, opracowywanie melodii czy zadania kontrapunktyczne. W jeszcze dalszej perspektywie, projekt mógłby zostać rozszerzony do szerszego zakresu teorii muzyki i zadań - chociażby z rytmiki czy kształcenia słuchu.

· **Tryb nauki krok po kroku:** Dodanie modułu wspomagającego naukę, który krok po kroku prowadziłby użytkownika przez proces rozwiązywania zadań. Dzięki temu początkujący uczniowie mogliby łatwiej zrozumieć reguły harmonii lub innych dziedzin muzyki oraz nauczyć się ich stosowania.

· **Integracja z platformami edukacyjnymi:** Wprowadzenie integracji z popularnymi platformami edukacyjnymi, takimi jak Moodle czy Google Classroom, pozwoliłoby na łatwą migrację uczniów oraz synchronizację ocen z systemami używanymi przez szkoły i placówki muzyczne.

Realizacja powyższych kierunków rozwoju pozwoliłaby uczynić projekt bardziej wszechstronnym i użytecznym zarówno dla uczniów, jak i nauczycieli, a także przyczyniłaby się do popularyzacji nauki teorii muzyki w przystępny sposób.

## 2.3 Podziękowania

Pragniemy wyrazić szczególne podziękowania naszemu opiekunowi za wsparcie na każdym etapie realizacji projektu. Podziękowania kierujemy również do osób zaangażowanych w testowanie naszej aplikacji i pomoc w identyfikacji obszarów wymagających poprawy: Uczniom i Nauczycielom Państwowej Szkoły Muzycznej Podwale, Studentom i Wykładowcom Akademii Muzycznej im. Karola Lipińskiego we Wrocławiu, a także wszystkim innym osobom, które przyczyniły się do rozwoju projektu.

## LITERATURA

[1] Musition. https://www.risingsoftware.com/musition. Last accessed 26.11.2024.

[2] Hooktheory. https://www.hooktheory.com/. Last accessed 26.11.2024.

[3] Microsoft. *Dokumentacja ASP.NET Core*, 2023. https://learn.microsoft.com/en-us/aspnet/core/?view=aspnetcore-8.0 Last accessed 26.11.2024.

[4] Microsoft. *Dokumentacja Azure*, 2024. https://learn.microsoft.com/pl-pl/azure/ Last accessed 26.11.2024.

[5] Microsoft. *Dokumentacja Entity Framework*, 2024. https://learn.microsoft.com/en-us/ef/ Last accessed 26.11.2024.

[6] Microsoft. *Dokumentacja Identity Framework*, 2024. https://learn.microsoft.com/en-us/aspnet/core/security/authentication/identity Last accessed 26.11.2024.

[7] Microsoft. *Dokumentacja MS SQL Server*, 2024. https://learn.microsoft.com/en-us/sql/sql-server/ Last accessed 26.11.2024.

[8] p5.js. *Dokumentacja p5.js*, 2024. https://p5js.org/reference/ Last accessed 26.11.2024.

[9] Kazimierz Sikorski. *Harmonia 1*. PWM/Pedagogika, 1948.