# OpenReview forum: "Platforma e-learningowa do przeprowadzania quizów z harmonii muzycznej z automatycznym sprawdzaniem i generowaniem rozwiązań"
_pwr.edu.pl/Wrocław_University_of_Science_and_Technology/2024/ZPI_Day — Wrocław University of Science and Technology 2024 ZPI Day Submission_

### Official Review · Reviewer_UDL8 · 2024-12-05
**Kahoot dla nauczycieli aranżowania harmonii muzycznej**

**Confidence:** 5
**Significance Of Results:** 5
**Overall Quality:** 5

**Compliance With Template:**

5: Very High Quality – The article contains all the required sections, which are written in a very detailed, clear, and error-free manner. The structure is professional and meets expectations, and the content adheres to the highest substantive and formal standards.

**Description Of Results:**

5: Very High Quality – The results are described in detail, clearly and comprehensively, supported by thorough evaluation, analysis, and convincing usage examples. The description meets the highest substantive standards.

**Feedback On Consistency:**

Opis projektu jest spójny i logiczny

**Potential For Development:**

Produkt jest niejako "na zamówienie" więc został już przetestowany w praktyce na Akademii Muzycznej. Oczywiście z tym wiążą się perspektywy dalszego rozwoju.

**Project Nature Evaluation:**

Projekt wykazuje cechy pracy inżynierskiej na poziomie użyteczności i rozwiązań technologicznych.
Pod względem problemu i podejścia do jego rozwiązania jest to produkt nowatorski.

Temat wyniknął z rzeczywistej potrzeby prowadzenia i auto-sprawdzania "kartkówek" z harmonii (aranżacji 4-głosu w kolejnych taktach utworu muzycznego używając zapisu nutowego) na Akademii Muzycznej. Można więc nawet powiedzieć, że jest to temat z biznesu :) Dostarczony produkt zapewnia nie tylko funkcje narzędzia testujące a ala Kahoot, ale przede wszystkim zapewnia edytor zapisu nutowego (+ odtwarzacz MIDI), porównywalny z profesjonalnymi narzędziami typu MuseScore.

**Technical Language Precision:**

5: Very High Quality – The language is entirely appropriate for a technical report. All terms are used correctly and precisely, and the style is professional, clear, and coherent, without any errors or ambiguities.

---

### Official Review · Reviewer_Upwj · 2024-12-06
**Przedstawione rozwiązanie trafnie identyfikuje lukę w narzędziach dostępnych do pracy w środowisku muzycznym. Jednocześnie mocną stroną projektu jest przetestowanie rozwiązania wśród osób, które mają zapotrzebowanie na takie narzędzia.**

**Confidence:** 2
**Significance Of Results:** 5
**Overall Quality:** 4

**Compliance With Template:**

5: Very High Quality – The article contains all the required sections, which are written in a very detailed, clear, and error-free manner. The structure is professional and meets expectations, and the content adheres to the highest substantive and formal standards.

**Description Of Results:**

5: Very High Quality – The results are described in detail, clearly and comprehensively, supported by thorough evaluation, analysis, and convincing usage examples. The description meets the highest substantive standards.

**Feedback On Consistency:**

Przedstawione rozwiązanie niewątpliwie zaspokaja bardzo istotny obszar środowiska muzycznego. Trafnie zidentyfikowano ograniczenia rozwiązania. Projekt czytelnie i klarownie przedstawia rozwiązanie natomiast w opisie brakowało informacji w jaki sposób użytkownik powinien wprowadzić zapis rozwiązania.

**Potential For Development:**

Pominięty został aspekt pierwszego wprowadzenia wszystkich zadań, by stworzyć bazę, która potem może być wykorzystywana przez szersze grono użytkowników. Warto również rozważyć moduł raportowy, który uwzględnia statystyki i wskazuje najczęściej popełniane błędy, podstawia zadania, które są problemowe dla użytkownika.

**Project Nature Evaluation:**

Przestawione rozwiązanie wykorzystuje odpowiednią technologię, co pozwala zakwalifikować go jako spełniające wymagania wobec prac inżynierskich.

**Technical Language Precision:**

4: High Quality – The language is appropriate for a technical report. Terminology is used correctly, and statements are precise, with only minor shortcomings that do not affect the overall clarity.

---

### Official Review · Reviewer_ZZYY · 2024-12-09
**A good, small, practical application**

**Confidence:** 4
**Significance Of Results:** 5
**Overall Quality:** 5

**Compliance With Template:**

4: High Quality – The article contains all the required sections, which are well-written and substantively correct, although minor errors or shortcomings may be present. The overall structure is clear and coherent.

**Description Of Results:**

5: Very High Quality – The results are described in detail, clearly and comprehensively, supported by thorough evaluation, analysis, and convincing usage examples. The description meets the highest substantive standards.

**Feedback On Consistency:**

The description is internally consisent and logical. It is quite clear what was done in the application and how it was tested. I would improve the structure slightly, but it is still readable as is.

**Potential For Development:**

The authors outline the necessary next steps in development of the application.

**Project Nature Evaluation:**

The practical tests of the application show that it is a complete engineering solution ready for an early phase of deployment.

**Technical Language Precision:**

4: High Quality – The language is appropriate for a technical report. Terminology is used correctly, and statements are precise, with only minor shortcomings that do not affect the overall clarity.

---

### Decision · Program_Chairs · 2024-12-10

Accept (Oral)